# Live Yeast or Live Yeast Combined with Zinc Oxide Enhanced Growth Performance, Antioxidative Capacity, Immunoglobulins and Gut Health in Nursery Pigs

**DOI:** 10.3390/ani11061626

**Published:** 2021-05-31

**Authors:** Shenfei Long, Tengfei He, Sung Woo Kim, Qinghui Shang, Tadele Kiros, Shad Uddin Mahfuz, Chunlin Wang, Xiangshu Piao

**Affiliations:** 1State Key Laboratory of Animal Nutrition, College of Animal Science and Technology, China Agricultural University, Beijing 100193, China; longshenfei@cau.edu.cn (S.L.); hetengfei@cau.edu.cn (T.H.); sqh123hxj456@163.com (Q.S.); wangchl@cau.edu.cn (C.W.); 2Department of Animal Science, North Carolina State University, Raleigh, NC 27695, USA; sungwoo_kim@ncsu.edu; 3Phileo by Lesaffre, 137 Rue Gabriel Péri, 59700 Marcq en Baroeul, France; t.kiros@phileo.lesaffre.com; 4Department of Animal Nutrition, Sylhet Agricultural University, Sylhet 3100, Bangladesh; shadmahfuz@yahoo.com

**Keywords:** antioxidant status, growth performance, immunoglobulins, live yeast, nursery pigs

## Abstract

**Simple Summary:**

The stress after weaning is the critical problem for nursery pigs. Although antibiotics and zinc oxide (ZnO) could be efficient methods to alleviate weaning stress, the abuse of them might be harmful for the environment. The current study found that live yeast (LY, *Saccharomyces cerevisiae* (strain CNCM I-4407, 10^10^ CFU/g)) or *S. cerevisiae* combined with ZnO replacing antibiotics and ZnO could enhance performance and reduce the diarrhea rate in nursery pigs via improving their nutrient utilization, antioxidant capacity, immunoglobulins, fecal volatile fatty acid composition, and microbiota community. The results of this study could be helpful for finding some novel strategies replacing in-feed antibiotics and ZnO to alleviate weaning stress in pigs.

**Abstract:**

This study aimed to investigate the effects of dietary LY or LY combined with ZnO supplementation on performance and gut health in nursery pigs. 192 Duroc × Landrace × Yorkshire piglets (weaned on d 32 of the age with 9.2 ± 1.7 kg BW) were allocated into four treatments with eight replicate pens, six piglets per pen. The treatments included a basal diet as control (CTR), an antibiotic plus ZnO diet (CTC-ZnO, basal diet + 75 mg/kg of chlortetracycline + ZnO (2000 mg/kg from d 1 to 14, 160 mg/kg from d 15 to 28)), a LY diet (LY, basal diet + 2 g/kg LY), and a LY plus ZnO diet (LY-ZnO, basal diet + 1 g/kg LY + ZnO). The results showed that pigs fed LY or LY-ZnO had increased (*p* < 0.05) average daily gain, serum IgA, IgG, superoxide dismutase, fecal butyric acid, and total volatile fatty acid concentrations, as well as decreased (*p* < 0.05) feed conversion ratio and diarrhea rate compared with CTR. In conclusion, pigs fed diets with LY or LY combined with ZnO had similar improvement to the use of antibiotics and ZnO in performance, antioxidant status, immunoglobulins, and gut health in nursery pigs.

## 1. Introduction

Weaning is a critical period in commercial pig production systems. Due to the nutritional, immunological, and psychological disruptions, nursery pigs may face with reduced feed intake, increased incidence of diarrhea, body weight loss, and higher mortality. Nutritional strategies were usually used to improve performance, and gut health in nursery pigs [1]. The use of different additives such as natural extracts are largely adopted in swine livestock [2], which have been widely studied for their promising antioxidant, anti-inflammatory and antibacterial properties [3]. Antibiotics have been used to enhance growth performance and to reduce the abundance of harmful bacteria in the intestine of nursery pigs. However, the overuse of antibiotics might lead to problems related with the bacterial resistance. Therefore, many countries (including China, European Union, Philippines, South Korea, United States of America and Vietnam) have banned the use of antibiotics in animal feeds. Zinc oxide (ZnO) is often used to decrease the occurrence of diarrhea in nursery pigs, whereas its excretion is a possible threat to the environment. Many international organizations, such as the European Food Safety Authority and World Health Organization, have recommended decreasing the use of copper and ZnO due to the suspected resistance to certain bacteria and risks associated with the impact of these heavy metals on the environment [2]. Probiotics can be used to replace ZnO [1] and antibiotics [4] in order to improve the structure and functions of the gastrointestinal tract. Moreover, probiotics could also be useful to improve intestinal development and immune functions in nursery pigs [5].

Yeast is one of the most commonly used probiotics replacing the use of antibiotics to reduce post-weaning diarrhea [6,7]. The yeast additives have been used to improve gut integrity in nursery pigs [8] and alleviate the negative effects on the growth and health caused by mycotoxins in nursery pigs [9,10]. Moreover, yeasts, especially live yeast (LY), could enhance antioxidative capacity and gut immunity by increasing secretory IgA level in mucosa and reducing harmful microbiota community in the intestine of nursery pigs [11]. Previous studies have shown that LY *Saccharomyces cerevisiae* (*S. cerevisiae*) could improve beneficial gut microbiota community in suckling or nursery piglets [12,13]. The yeast extract complex derived from the culture of *Kluyveromyces maxianus* and *S. cerevisiae* could also improve nutrient digestibility in nursery pigs [14]. Moreover, *S. cerevisiae* have been widely used to reduce diarrhea and to modulate gut health in animals and humans [15,16], which were also used to improve the immunity, gut development, microbiota community, and thus alleviate weaning stress in nursery pigs [7,17,18].

Previously, it was found that *S. cerevisiae* could replace chlortetracycline on improving immune function, antioxidant capacity, nutrient digestibility, gut morphology and performance in broilers [19]. Supplemental effects of *S. cerevisiae* might be further enhanced with a combinational use of ZnO. Based on previous findings, it is hypothesized that dietary supplementation with LY or LY combined with ZnO could improve performance, nutrient digestibility, antioxidative status, immune function and gut health in nursery pigs. Therefore, the objective of the current study was to test this hypothesis and find effective feed additives combination to alleviate weaning stress in pigs.

## 2. Materials and Methods

The animal procedures in this study were approved by the Institutional Animal Care and Use Committee of China Agricultural University (Beijing, China) (No. AW52301202-1-1). The trial was conducted at the Feng Ning Swine Research Unit of China Agricultural University (Hebei, Chengde, China).

### 2.1. Experimental Products

The LY (*S. cerevisiae* (strain CNCM I-4407), 10^10^ CFU/g) was obtained from Phileo by Lesaffre (Marcq-en-Baroeul, France). The ZnO and chlortetracycline were provided by Beijing Tonglixingke Agricultural Science and Technology Co., Ltd. (Beijing, China).

### 2.2. Experimental Animals, Management, and Design

A total of 192 piglets (Duroc × (Landrace × Yorkshire), weaned on d 32 of the age, initial BW was 9.19 ± 1.68 kg) were allocated into 4 dietary treatments with 8 replicate pens, 6 piglets (3 barrows and 3 gilts) per pen in a complete random design. The trial was divided into two experimental phases: phase 1 (d 1–14) and phase 2 (d 15–28). The treatments included a corn-soybean meal basal diet as control (CTR), an antibiotic plus ZnO diet (CTC-ZnO, basal diet + 75 mg/kg of chlortetracycline + ZnO (2000 mg/kg in phase 1, 160 mg/kg in phase 2)), a LY diet (LY, basal diet + 2 g/kg LY), and a LY plus ZnO diet (LY-ZnO, basal diet + 1 g/kg LY + ZnO (2000 mg/kg in phase 1, 160 mg/kg in phase 2)). All the treatments were supplemented in both phases 1 and 2. Chromic oxide (2.5 g/kg) was used as an indigestible marker, which was supplemented for all the experimental treatments. The nutrient level in basal diet met the requirements recommended by NRC [20] (Table 1).

### 2.3. Data Recording, Sample Collection and Analysis

The relative humidity in the nursery room was maintained at 60–70%, while the temperature was maintained at 24 ± 2 °C. Pigs were raised in 1.5 × 1.5 m^2^ pens equipped with adjustable feeders, duckbill drinkers and plastic slatted floors. Pigs had free access to water and feed (in a mash form). All the feed and pigs were weighed on d 1, 7, 14 and 28 to calculate the average daily gain (ADG), average daily feed intake (ADFI) and feed conversation ratio (FCR, ADFI/ADG), respectively. A scoring system was applied to indicate the presence and severity of diarrhea as follows: 1 = hard feces; 2 = slightly soft feces; 3 = soft, partially formed feces; 4 = loose, semiliquid feces; and 5 = watery, mucous-like feces. When the average score was over 3, pigs were identified as having diarrhea. Diarrhea rate was calculated as following: diarrhea rate (%) = (the diarrhea days × the number of diarrhea pigs)/(the total experiment days × the total number of pigs) × 100 [21].

About 1 kg fresh fecal samples from all piglets in each pen (*n* = 8) were collected using the grab sample technique according to Long et al. [21] from d 12 to 14 and d 26 to 28, and these fresh fecal samples were dried for 3 days in a 65 °C oven. The dried fecal samples and feed were ground and passed through a 1-mm sieve to measure gross energy (GE), dry matter (DM), ash, organic matter (DM-ash, OM), crude protein (CP), ether extract (EE) and chromium (Cr) following the methods of AOAC [22]. The GE content in the dried fecal samples and feed was measure by an automatic isoperibolic oxygen bomb calorimeter (Parr 1281 Automatic Energy Analyzer, Moline, IL, USA). According to Van Soest et al. [23], the fiber analyzer (Ankom Technology, Macedon, NY, USA) was used to measure the level of acid detergent fiber (ADF) and neutral detergent fiber (NDF). The Gr content of the dried fecal samples and feed was measured by the atomic absorption spectrophotometer (Z-5000; Hitachi, Tokyo, Japan). The apparent total tract digestibility (ATTD) of nutrients was measured following the formula: ATTD_nutrient_ = 1 − (nutrient_feces_ × Cr_diet_)/(nutrient_diet_ × Cr_feces_) [21].

On the morning of d 7, 14, 21, and 28, barrows weighing near the average BW in each pen (*n* = 8) was used for the collection of blood samples (about 8 mL) via the jugular vein into vacutainer (Becton Dickinson Vacutainer Systems, Franklin Lakes, NJ, USA). The blood samples were centrifuged at 3000× *g* and 4 °C for 15 min to get the serum samples and stored at −20 °C until analysis. An ELISA kit (IgG, IgM and IgA quantitation kit; Bethyl Laboratories, Inc., Montgomery, TX, USA) was used for the measurement of the serum immunoglobulins levels. The malondialdehyde (MDA), superoxide dismutase (SOD), total antioxidant capacity (T-AOC), glutathione peroxidase (GSH-Px) and catalase (CAT) levels in serum were measured by a spectrophotometer (LengGuang SFZ1606017568, Shanghai, China) according to the instructions of the corresponding reagent kits (Nanjing Jiancheng Institute of Bioengineering, Nanjing, China).

The fresh fecal samples of barrows weighing near the average BW in each pen (n = 8) on d 28 were used for the measurement of volatile fatty acids (VFA) contents and microbiota community. The VFA contents in fecal samples were measured by a Hewlett Packard 5890 gas chromatograph (HP, Avondale, PA, USA) following the procedure mentioned by Long et al. [21]. About 1.5 g fresh fecal sample was weighed into a centrifuge tube, mixed with 1.5 mL sterile water, and centrifuged at 15,000× *g* and 4 °C for 15 min. The supernatant (1 mL) was transferred into a Gas Chromatograph sample bottle and 200 μL meta-phosphoric acid was used to mix the samples. Then, the samples were placed for 30 min in ice, and centrifuged at 15,000× *g* and 4 °C for 15 min.

The total genomic DNA of bacteria from fecal samples was determined by DNA Kit (omega bio TEK, Norcross, GA, USA). The concentration and purity of DNA were detected by NanoDrop 2000 Spectrophotometer (Thermo Scientific, DE, Wilmington, USA). The 16S rRNA gene of the V3-V4 region was amplified by using a pair of primers (338F ACTCCTACGGGAGGCAGCAG and 806R GGACTACHVGGGTWTCTAAT). The amplified product was recovered and purified by axyprep DNA gel extraction kit (Axygen Biosciences, Union City, CA, USA), and the amplified product was purified by qubit 2.0 fluorescence meter (Thermo Fisher Scientific, Waltham, MA, USA) to be quantified to a uniform concentration. The amplified fragment was established and sequenced on the Illumina hiseq pe300 platform (Illumina, San Diego, CA, USA). The matched terminal reading code was 300 base pair (BP). The original sequencing sequence was quality controlled by Trimmomatic (version 3.29). Flash (version 1.2.7) software was spliced, and UPARSE (version 7.1) software was used to cluster operational taxonomic unit (OTU). The similarity threshold was 0.97, and OTU was classified and analyzed by Ribosomal Database Project (RDP) database (https://rdp.cme.msu.edu/) [24]. The Quantitative Insights Into Microbial Ecology (QIIME) was used for the analysis of to the α-diversity [25]. The relative abundance of bacteria is expressed as a percentage.

### 2.4. Statistical Analysis

A randomized complete block design was used in this study with sex and initial BW as blocking criteria. Mixed procedure of SAS (version 9.2, 2008) [26] was used for statistical analysis. Dietary treatment was the fixed effect, while the block was the random effect. For performance, diarrhea rate and nutrient digestibility, individual pen was used as statistical unit, while for other data, the individual pig was used as statistical unit. The difference of diarrhea rate was analyzed by chi-square contingency test, while the statistical differences of other data except for the microbiota were analyzed via the Student–Neuman–Keul’s multiple range tests. Differences of microbiota abundance in feces were analyzed using a Kruskal–Wallis rank sum test. Significant difference was defined as *p* ≤ 0.05, and a trend of difference was determined as 0.05 < *p* ≤ 0.10.

## 3. Results

### 3.1. Performance and Diarrhea Rate

Pigs fed LY or LY-ZnO showed increased (*p* < 0.05) ADG from d 1 to 7, d 8 to 14, d 1 to 14 and d 1 to 28 compared with CTR, and enhanced (*p* < 0.05) ADG from d 1 to 7 compared with CTC-ZnO. Pigs fed LY-ZnO had increased (*p* < 0.05) ADFI from d 1 to 7 compared with CTR and CTC-ZnO. Pigs fed LY-ZnO also had decreased (*p* < 0.05) FCR from d 8 to 14 and d 1 to 14 compared with CTR. Pigs fed LY had decreased (*p* < 0.05) FCR from d 1 to 7, d 1 to 14, and d 1 to 28 compared with CTR, and lower (*p* < 0.05) FCR from d 1 to 7 compared with CTC-ZnO (Table 2).

Pigs fed CTR showed a higher (*p* < 0.05) diarrhea rate from d 8 to 14, d 1 to 14 and d 1 to 28 compared with other treatments. From d 8 to 14, the diarrhea rate was decreased (*p* < 0.05) in pigs fed LY-ZnO compared with CTC-ZnO and LY (Table 2).

### 3.2. The ATTD of Nutrients

In phase 1, pigs fed LY or LY-ZnO had increased (*p* < 0.05) ATTD of GE, CP and EE compared with CTR. In phase 2, pigs fed LY showed increased (*p* < 0.05) ATTD of CP, while pigs fed LY or LY-ZnO also had higher (*p* < 0.05) ATTD of OM and NDF compared with CTR (Table 3).

### 3.3. Immunoglobulins Levels in Serum

On d 7 and 14, the concentration of both IgA and IgG in serum of pigs fed LY or LY-ZnO were increased (*p* < 0.05) compared with CTR or CTC-ZnO (Table 4).

### 3.4. Antioxidant Status in Serum

On d 28, the serum MDA level in pigs fed CTR was higher (*p* < 0.05) compared with other treatments. On d 7, 14, 21, and 28, the serum SOD content in pigs fed LY-ZnO was increased (*p* < 0.05) compared with CTR. On d 7, 14, and 28, the serum SOD content in pigs fed LY was increased (*p* < 0.05) compared with CTR. On d 7 and 21, the serum SOD content in pigs fed CTC-ZnO was increased (*p* < 0.05) compared with CTR. On d 7, the serum T-AOC level was reduced (*p* < 0.05) in pigs fed CTC-ZnO compared with CTR and LY. On d 21, the serum CAT level in pigs fed CTR was lower (*p* < 0.05) compared with other treatments (Table 5).

### 3.5. Concentrations of VFA in Feces

On d 14, the concentration of propionic acid, butyric acid and total VFA in feces of pigs fed LY or LY-ZnO were increased (*p* < 0.05) compared with CTR.

On d 28, the concentrations of butyric acid and total VFA in feces of pigs fed LY or LY-ZnO were increased (*p* < 0.05) compared with CTR and CTC-ZnO (Table 6).

### 3.6. Fecal Microbiota Composition

There was no significant difference of α-diversity of microbiota in feces among treatment (Table 7. We also found the relative abundance of Firmicutes on phylum level in feces of pigs fed LY were higher (*p* < 0.05) than pigs fed LY-ZnO. Moreover, the relative abundance of *Faecalibacterium*, *Prevotellaceae_NK3B31_group*, *norank_f__Prevotellaceae*, and *Anaerovibrio* at the genus level in feces of pigs fed CTR were reduced (*p* < 0.05) compared with pigs fed LY or LY-ZnO (Table 8).

## 4. Discussion

The performance of piglets in the first week after weaning might play an important role in the whole production performance. Currently, antibiotics have been widely used to enhance performance and alleviate weaning stress in pigs, while the abuse of antibiotics might do harm to the environment and human health [21]. High dose of inorganic Zn (2000–4000 mg/kg of Zn as ZnO) has been widely used in the diets of piglets in the swine industry, due to its effects on increasing growth performance and alleviating post-weaning diarrhea [27]. An increased use of zinc oxide at a pharmacological level (ZnO, 2000–3000 mg/kg) was also observed as an alternative to antibiotics. However, the widespread use of pharmacological levels of ZnO has raised concerns related to environmental issues and the potential increase in the prevalence of antibiotic resistant bacteria [2]. However, the overuse of ZnO could pollute the environment and could lead to increased antibiotic resistant bacteria since the zinc could not to be fully utilized by pigs. The current study aimed to investigate the effect of LY or LY combined with ZnO as antibiotics and ZnO substitute on performance and gut health in nursery pigs. Mathew et al. [28] reported that yeast extract could improve ADG and reduce FCR in nursery pigs, indicating that yeast could be probiotic for enhancing performance in pigs. According to the current result, we found pigs fed LY or LY-ZnO had similar effect as antibiotics and ZnO on increasing ADG and decreasing FCR in the first, second, and fourth weeks compared with CTR. One of the reason for this finding might be that *S. cerevisiae* could improve nutrient utilization and gut morphology [19]. Another possible mechanism of this finding might be that the effect of β-glucan and mannan oligosaccharides in the cell wall of LY (*S. cerevisiae*) could improve immunity and small intestinal development in nursery pigs [17,29,30]. In the present study, we also found pigs fed LY-ZnO had increased ADFI in the first week compared with CTR and CTC-ZnO, which might be due to that LY could improve the flavor of diet, enhance intestinal peristalsis, and facilitate digestion and absorption in nursery pigs [31].

The incidence of diarrhea during weaning is a great economic issue in production stage. Nutritional strategies (such as dietary supplement of probiotics) can be used to promote intestinal development of nursery pigs and to improve diarrhea caused by dysbacteriosis of piglets. Pan et al. [5] reported that probiotics had significant effects on the prevention and treatment of diarrhea in nursery pigs. In the current study, pigs fed a CTR diet showed higher diarrhea rate in phase 1 and overall compared with other treatments, which agreed with the study of Shen et al. [6], who reported that yeast could reduce the incidence of diarrhea in nursery pigs. One of the possible reasons for the current finding might be that yeast could help to increase beneficial microbial populations and decrease the populations of pathogens in piglets [32]. Moreover, Kim et al. [33] also reported that the nucleotide-rich yeast extract could improve gut health, which could reduce the diarrhea in nursery pigs

After weaning, the nutrient utilization was decreased since digestive organs were not fully developed in nursery pigs. In the current study, we found pigs fed LY or LY-ZnO showed increased ATTD of GE, CP, and EE compared with CTR in phase 1, indicating that LY or LY-ZnO could help improve the development of digestive organs and nutrient utilization. This finding was partly agreed with Shen et al. [6], who reported yeast culture could enhance the digestibility of DM, CP and GE, and thus decrease the FCR in nursery pigs. Moreover, the current study showed pigs fed LY had increased ATTD of CP, while pigs fed LY or LY-ZnO also showed increased ATTD of OM and NDF compared with CTR in phase 2. The reason for this finding might be that the *S. cerevisiae* could enhance the development of small intestine [19].

For nursery pigs, the immunoglobulins are important for the development of immune system. Shen et al. [34] reported that maternal yeast supplementation could enhance the IgG contents in sow milk, which could help to increase litter weight and weaning piglet‘s weight [34,35], indicating yeast had positive effects on increasing immunoglobulins levels in animals. The current study found that the concentration of both IgA and IgG in serum of pigs fed LY or LY-ZnO were increased compared with CTR and CTC-ZnO on d 7 and d 14. The mannanoligosaccharides and β-glucans were the main components of LY cell wall and LY [19]. The current finding was in agreement with Wang et al. [36] and He et al. [19], who also reported that the Ig levels were enhanced by *S. cerevisiae*, which might be due to the β-glucan in LY also has anti-inflammatory and immune regulatory functions [37]. Besides, the nucleotide in yeast might also help to enhance immune response via increasing inflammatory cytokines in ileum and modulate proliferation of beneficial gut bacteria in nursery pigs [38].

After weaning, pigs were faced with severe oxidative stress, which might cause lipid peroxidation and disruption of DNA replication and protein synthesis. Lipid peroxidation transforms reactive oxygen species into active chemicals, which could cause cell metabolism and dysfunction, or even death. The MDA was a biomarker, reflecting the rate and intensity of lipid peroxidation in tissue. Therefore, the measurement of MDA could reflect the degree of lipid peroxidation and indirectly measure the degree of cell damage. In the current study, the serum MDA level in pigs fed CTR was higher compared with other treatments on d 28, indicating LY could alleviate the lipid peroxidation in nursery pigs. He et al. [19] also reported that *S. cerevisiae* could improve serum SOD level and decrease MDA level in broilers, which demonstrated that *S. cerevisiae* could potentially enhance antioxidant capacity via improving antioxidant enzyme activities. The SOD was one of the antioxidant enzymes which help alleviate oxidative stress after weaning. In the current study, the serum SOD content in pigs fed LY was increased on d 7, 14, and 28 compared with CTR, which might be the polysaccharide in LY could enhance the antioxidant activity [39]. The mechanism of LY on alleviating oxidative stress (reducing MDA level) might be related to the beneficial effect of β-glucan in LY [40]. Moreover, the β-glucan in LY could also enhance antioxidant capacity and reduce oxidative damage of lymphocytes through different enzymatic and non-enzymatic systems [37].

In order to figure out the possible mechanism of LY on modulating gut health in nursery pigs, we investigated the fecal microbiota community and fecal VFA contents. The VFA produced by microbial fermentation of non-digestible carbohydrates in large intestine is beneficial to animals [41]. The addition of yeast extract to diet could utilize carbohydrate to produce VFA, which had a positive effect on the growth epithelial cells in pigs [42]. In the current study, we found that the concentration of propionic acid, butyric acid and TVFA in feces of pigs fed LY or LY-ZnO were increased on d 14 compared with CTR. On d 28, the concentrations of butyric acid and TVFA in feces of pigs fed LY or LY-ZnO were increased compared with CTR and CTC-ZnO. These results are in agreement with a previous study which reported that fecal samples from yeast fed sows increase fermentation and VFA production when incubated in vitro with different sources of fiber [43]. The VFA are the gut microbiota metabolites, which play an important role on solving gut diseases. The VFA produced by bacterial fermentation was an important component of colonic contents, which could be rapidly absorbed by colonic epithelial cells, participate in metabolism and supply energy [41], while the organic acids in large intestine could also enhance the immunity and regulate the micro-ecological environment in the digestive tract of piglets [21,44]. Therefore, the trend for the increased TVFA content in the large intestine was beneficial for the immunity and gut health of nursery pigs in the current study.

In this study, we found the relative abundance of Firmicutes on phylum level in feces of pigs fed LY were higher than pigs fed LY-ZnO. While Kiros et al. [13] reported that the enhanced Firmicutes family by *S. cerevisiae* might contribute to regulate intestinal homeostasis and to improve the performance of piglets. Therefore, LY alone might be more effective in regulate intestinal homeostasis than LY combined with ZnO, but its mechanism still remained to be investigated. It has been reported that the LY *S. cerevisiae* was effective in alleviating the negative effect by *E. coli* on jejunal mucosa, which indicated that *S. cerevisiae* could modulate the gut microbiota community in pigs [45]. In this study, we found the relative abundances of *Faecalibacterium*, *Prevotellaceae_NK3B31_group*, *norank_f__Prevotellaceae*, and *Anaerovibrio* at the genus level were higher in feces of pigs LY or LY-ZnO compared with CTR. Villot et al. [46] reported that *S. cerevisiae* could increase the level *Lactobacillus* and tended to have higher *Faecalibacterium prausnitzii* content in the jejunum, which might help stimulate IgA in the gut of newborn calves. Besides, dietary supplementation with LY *S. cerevisiae* could also improve the relative abundance of *Prevotella* in cecum, which was positively correlated with the increased ADG in piglets [12]. The positive correlation between *Prevotella* abundancy and the piglets’ growth performance has been attributed to its ability to process complex dietary saccharides of the diet and favoring monosaccharide uptake by the host [47]. The *Prevotella* could also use monosaccharide to produce VFA [48], which was conducive to the digestion and absorption of nutrients in nursery pigs. The production of VFA was one of the mechanisms by which intestinal microbiota could influence and promote the host metabolism and physiology [49] and might explain the positive correlation between *Prevotellaceae_NK3B31_group, norank_f__Prevotellaceae,* and performance by LY or LY-ZnO observed in our study.

## 5. Conclusions

The current study demonstrated that LY (*S. cerevisiae* (strain CNCM I-4407)) or LY combined with ZnO as feed additives could be used to alleviate weaning stress in nursery pigs. Moreover, LY or LY combined with ZnO also had positive effects on growth performance via improving nutrient digestibility, serum immunoglobulin, antioxidant status, and intestinal health (fecal VFA contents and microbiota community) in nursery pigs.

## Figures and Tables

**Table 1 animals-11-01626-t001:** Ingredient composition and nutrient levels of diets (as-fed basis, %).

Ingredients	Phase 1 (d 1 to 14)	Phase 2 (d 15 to 28)
Corn	58.52	63.22
Soybean meal	14.16	15.28
Extruded soybean	10.00	8.00
Whey powder, 3.8%	4.00	4.00
Prosaf ^1^	4.00	4.00
Soy oil	3.00	2.28
Fish meal	3.00	0.00
Limestone	1.00	0.80
Dicalcium phosphate	0.75	0.65
L-lysine HCl	0.34	0.45
Salt	0.30	0.30
DL-Methionine	0.09	0.08
L-Threonine	0.07	0.15
L-Tryptophan	0.02	0.04
Chromic oxide	0.25	0.25
Vitamin-mineral premix ^2^	0.50	0.50
Calculated nutrient level		
Digestible energy, kcal/kg	3542	3490
Crude protein	20.00	18.00
Calcium	0.80	0.70
Digestible phosphorus	0.40	0.33
Standard ileal digestible lysine	1.35	1.23
Standard ileal digestible methionine	0.39	0.36
Standard ileal digestible threonine	0.79	0.73
Standard ileal digestible tryptophan	0.22	0.20
Analyzed nutrient level		
Crude protein	20.26	18.29
Calcium	0.82	0.73
Phosphorus	0.71	0.62
Lysine	1.37	1.24
Methionine	0.42	0.37
Threonine	0.82	0.77

Note: ^1^ Prosaf was provided by Phileo Lesaffre Animal Care. The product “Prosaf” is a yeast extract obtained by autolysis of proprietary *Saccharomyces cerevisiae* baker’s yeast strains. The product is a sustainable source of highly digestible and palatable protein and peptides rich in essential amino acids and glutamate. ^2^ Premix provided the following per kg of feed: vitamin A, 12,000 IU; vitamin D_3_, 2500 IU; vitamin E, 30 IU; vitamin K_3_, 30 mg; vitamin B_12_, 12 μg; riboflavin, 4 mg; pantothenic acid, 15 mg; niacin, 40 mg; choline chloride, 400 mg; folic acid, 0.7 mg; vitamin B_1_, 1.5 mg; vitamin B_6_, 3 mg; biotin, 0.1 mg; Mn, 40 mg; Fe, 90 mg; Zn, 100 mg; Cu, 8.8 mg; I, 0.35 mg; Se, 0.3 mg.

**Table 2 animals-11-01626-t002:** Effects of live yeast or live yeast combined with zinc oxide on performance and diarrhea rate of nursery pigs.

Items	CTR ^1^	CTC-ZnO ^1^	LY ^1^	LY-ZnO ^1^	SEM	*p*-Value
d 1 BW, kg	9.17	9.20	9.14	9.19	0.183	0.99
ADG 2, g						
d 1 to 7	425 ^b^	435 ^b^	510 ^a^	502 ^a^	14.723	<0.01
d 8 to 14	469 ^b^	548 ^a^	539 ^a^	565 ^a^	19.558	0.01
d 15 to 21	555	544	605	573	23.311	0.30
d 22 to 28	585	585	620	613	12.230	0.11
d 1 to 14	447 ^b^	491 ^ab^	525 ^a^	534 ^a^	15.078	<0.01
d 15 to 28	570	564	612	593	15.548	0.14
d 1 to 28	508 ^b^	528 ^ab^	569 ^a^	563 ^a^	14.320	0.02
ADFI 2, g						
d 1 to 7	724 ^b^	711 ^b^	772 ^ab^	805 ^a^	22.312	0.03
d 8 to 14	795	852	843	858	34.364	0.57
d 15 to 21	931	914	1013	995	37.361	0.21
d 22 to 28	1085	1062	1153	1116	30.790	0.21
d 1 to 14	760	781	807	831	22.938	0.17
d 15 to 28	1008	988	1083	1055	26.816	0.08
d 1 to 28	884	885	945	943	20.862	0.07
FCR 2, g/g						
d 1 to 7	1.70 ^a^	1.65 ^a^	1.52 ^b^	1.60 ^a^	0.030	<0.01
d 8 to 14	1.69 ^a^	1.56 ^ab^	1.57 ^ab^	1.52 ^b^	0.036	0.02
d 15 to 21	1.68	1.69	1.67	1.74	0.040	0.68
d 22 to 28	1.86	1.82	1.86	1.82	0.037	0.81
d 1 to 14	1.70 ^a^	1.60 ^ab^	1.54 ^b^	1.56 ^b^	0.025	<0.01
d 15 to 28	1.77	1.76	1.77	1.78	0.026	0.94
d 1 to 28	1.74 ^a^	1.68 ^ab^	1.66 ^b^	1.68 ^ab^	0.020	<0.01
Diarrhea rate, %						
d 1 to 7	14.3 ^a^	3.6 ^b^	7.1 ^ab^	6.5 ^ab^	2.406	0.03
d 8 to 14	29.0 ^a^	15.0 ^b^	17.9 ^b^	5.8^c^	1.443	<0.01
d 15 to 21	12.1 ^a^	9.3 ^ab^	2.2 ^b^	5.3 ^ab^	2.342	0.03
d 22 to 28	2.3	1.5	0.0	0.0	0.645	0.06
d 1 to 14	21.7 ^a^	9.3 ^bc^	12.5 ^b^	6.1^c^	1.511	<0.01
d 15 to 28	7.2 ^a^	5.4 ^ab^	1.1 ^b^	2.6 ^ab^	1.370	0.02
d 1 to 28	14.4 ^a^	7.3 ^b^	6.8 ^b^	4.4 ^b^	1.237	<0.01

Note: SEM is standard error of mean. *n* = 8 indicating the data was analyzed in 8 pens per treatment. ^a–c^ Within a row, different superscripts mean significant difference (*p* < 0.05). ^1^ CTR: control; CTC-ZnO: chlortetracycline + zinc oxide; LY: live yeast; LY-ZnO: live yeast + zinc oxide. ^2^ ADG, average daily gain; ADFI, average daily feed intake; FCR, feed conversion ratio.

**Table 3 animals-11-01626-t003:** Effects of live yeast or live yeast combined with zinc oxide on apparent total tract digestibility of nutrient in nursery pigs.

Items (%)	CTR ^1^	CTC-ZnO ^1^	LY ^1^	LY-ZnO ^1^	SEM	*p*-Value
Gross energy						
d 14	81.27 ^b^	82.28 ^ab^	83.55 ^a^	83.28 ^a^	0.407	<0.01
d 28	83.39	83.17	84.21	83.61	0.406	0.33
Dry matter						
d 14	82.30	82.42	82.88	82.52	0.472	0.84
d 28	84.06	84.51	85.00	84.61	0.391	0.44
Organic matter						
d 14	83.53	83.70	84.10	84.84	0.560	0.38
d 28	83.31 ^b^	84.10 ^b^	86.43 ^a^	86.13 ^a^	0.370	<0.01
Crude protein						
d 14	76.48 ^b^	77.28 ^ab^	78.55 ^a^	78.16 ^a^	0.390	0.01
d 28	77.31 ^b^	78.12 ^ab^	79.39 ^a^	79.06 ^ab^	0.419	0.01
Ether extract						
d 14	64.27 ^b^	65.28 ^ab^	66.55 ^a^	66.16 ^a^	0.475	0.01
d 28	66.22	66.00	67.30	66.73	0.675	0.54
Neutral detergent fiber						
d 14	48.42	49.28	49.91	49.56	0.479	0.18
d 28	48.18 ^b^	49.84 ^a^	51.22 ^a^	51.22 ^a^	0.453	<0.01
Acid detergent fiber						
d 14	42.17	42.28	42.91	42.56	0.364	0.49
d 28	43.23	43.57	44.10	43.71	0.322	0.34

Note: SEM is standard error of mean. *n* = 8 indicating the data was analyzed in 8 pens per treatment. ^a,b^ Within a row, different superscripts mean significant difference (*p* < 0.05). ^1^ CTR: control; CTC-ZnO: chlortetracycline + zinc oxide; LY: live yeast; LY-ZnO: live yeast + zinc oxide.

**Table 4 animals-11-01626-t004:** Effects of live yeast or live yeast combined with zinc oxide on serum immunoglobulins in nursery pigs.

Items (ug/mL)	CTR ^1^	CTC-ZnO ^1^	LY ^1^	LY-ZnO ^1^	SEM	*p*-Value
IgA						
d 7	61.12 ^b^	64.05 ^b^	92.42 ^a^	91.19 ^a^	2.664	<0.01
d 14	54.08 ^b^	46.47 ^c^	90.31 ^a^	89.83 ^a^	1.959	<0.01
d 21	90.83	92.32	93.55	92.64	1.522	0.66
d 28	86.84	89.46	90.63	97.36	4.634	0.10
IgG						
d 7	158.63 ^c^	175.04 ^b^	270.97 ^a^	264.02 ^a^	4.551	<0.01
d 14	188.05 ^b^	201.27 ^b^	269.29 ^a^	266.66 ^a^	5.267	<0.01
d 21	275.77	269.72	271.28	276.69	2.984	0.31
d 28	264.55	276.22	289.38	282.22	9.812	0.06
IgM						
d 7	18.60	18.21	20.81	19.60	1.686	0.70
d 14	18.09	22.13	22.28	21.32	2.039	0.45
d 21	20.76	20.84	20.61	19.59	2.069	0.91
d 28	20.21	20.35	20.13	22.63	2.312	0.85

Note: SEM is standard error of mean. *n* = 8 indicating the data was analyzed in 8 pigs per treatment. ^a–c^ Within a row, different superscripts mean significant difference (*p* < 0.05). ^1^ CTR: control; CTC-ZnO: chlortetracycline + zinc oxide; LY: live yeast; LY-ZnO: live yeast + zinc oxide.

**Table 5 animals-11-01626-t005:** Effects of live yeast or live yeast combined with zinc oxide on serum antioxidant status of nursery pigs.

Items ^2^	CTR ^1^	CTC-ZnO ^1^	LY ^1^	LY-ZnO ^1^	SEM	*p*-Value
MDA, nmol/mL						
d 7	2.40	2.23	2.14	2.06	0.256	0.82
d 14	2.16	1.91	2.11	2.18	0.211	0.80
d 21	2.61	2.95	2.58	2.51	0.208	0.49
d 28	4.36^a^	3.37 ^b^	2.45 ^c^	2.25 ^c^	0.246	<0.01
SOD, U/mL						
d 7	69.71 ^c^	74.92 ^b^	79.16 ^a^	75.71 ^b^	1.073	<0.01
d 14	72.62 ^b^	78.93 ^ab^	77.93 ^a^	81.50 ^a^	1.177	<0.01
d 21	77.68 ^b^	98.91^a^	78.84 ^b^	99.65 ^a^	0.925	<0.01
d 28	82.35 ^c^	82.41 ^c^	88.53 ^b^	96.92 ^a^	1.114	<0.01
T-AOC, U/mL						
d 7	2.59 ^a^	2.36 ^b^	2.56 ^a^	2.41 ^ab^	0.060	0.04
d 14	2.56	2.46	2.44	2.45	0.043	0.21
d 21	2.42	2.33	2.48	2.59	0.107	0.38
d 28	2.43	2.40	2.33	2.38	0.027	0.12
GSH-Px, nmol/L						
d 7	20.88	19.22	21.00	19.64	0.560	0.09
d 14	20.76	22.00	19.63	19.79	1.070	0.40
d 21	20.41	20.19	20.82	20.03	1.417	0.98
d 28	19.53	20.23	19.24	19.76	0.711	0.80
CAT, U/mL						
d 7	8.23	10.47	10.03	10.59	0.693	0.09
d 14	8.22	8.54	8.60	8.63	1.615	0.70
d 21	11.05 ^b^	12.57 ^a^	12.88 ^a^	12.67 ^a^	0.417	0.02
d 28	11.29	9.90	11.30	10.87	0.904	0.65

Note: SEM is standard error of mean. *n* = 8 indicating the data was analyzed in 8 pigs per treatment. ^a–c^ Within a row, different superscripts mean significant difference (*p* < 0.05). ^1^ CTR: control; CTC-ZnO: chlortetracycline + zinc oxide; LY: live yeast; LY-ZnO: live yeast + zinc oxide. ^2^ MDA: malondialdehyde; SOD: superoxide dismutase; T-AOC: total antioxidant capacity; GSH-Px: glutathione peroxidase; CAT: catalase.

**Table 6 animals-11-01626-t006:** Effects of live yeast or live yeast combined with zinc oxide on volatile fatty acids concentration in feces on d 28 of nursery pigs.

Items (mg/g)	CTR ^1^	CTC-ZnO ^1^	LY ^1^	LY-ZnO ^1^	SEM	*p*-Value
Acetic acid						
d 14	1.55	1.51	1.54	1.66	0.109	0.78
d 28	1.62	1.55	1.74	1.97	0.112	0.07
Propionic acid						
d 14	0.71 ^b^	0.82 ^ab^	0.92 ^a^	0.99 ^a^	0.057	0.02
d 28	1.00	1.01	1.12	1.20	0.072	0.20
Butyric acid						
d 14	0.70 ^b^	0.77 ^b^	0.95 ^a^	0.97 ^a^	0.046	<0.01
d 28	0.73 ^b^	0.81 ^b^	0.95 ^b^	1.16 ^a^	0.062	<0.01
Isovaleric acid						
d 14	0.24	0.26	0.33	0.29	0.024	0.08
d 28	0.31	0.31	0.37	0.30	0.024	0.15
Valeric acid						
d 14	0.27	0.25	0.29	0.29	0.011	0.06
d 28	0.26	0.25	0.29	0.29	0.013	0.09
Total volatile fatty acid						
d 14	3.62 ^b^	3.78 ^b^	4.17 ^a^	4.32 ^a^	0.131	<0.01
d 28	4.05 ^b^	4.07 ^b^	4.64 ^ab^	5.09 ^a^	0.174	<0.01

Note: SEM is standard error of mean. *n* = 8 indicating the data was analyzed in 8 pigs per treatment. ^a,b^ Within a row, different superscripts mean significant difference (*p* < 0.05). ^1^ CTR: control; CTC-ZnO: chlortetracycline + zinc oxide; LY: live yeast; LY-ZnO: live yeast + zinc oxide.

**Table 7 animals-11-01626-t007:** Effects of live yeast or live yeast combined with zinc oxide on α-diversity of microbiota in feces on d 28 of nursery pigs.

Items	CTR ^1^	CTC-ZnO ^1^	LY ^1^	LY-ZnO ^1^	SEM	*p*-Value
Sobs	365.50	288.33	367.00	345.00	29.143	0.26
Shannon	3.68	3.60	4.02	3.95	0.152	0.31
Simpson	0.06	0.07	0.04	0.05	0.012	0.08
Ace	401.57	325.88	414.37	383.42	29.813	0.16
Chao	408.10	325.33	423.61	391.67	28.742	0.10

Note: SEM is standard error of mean. *n* = 8 indicating the data was analyzed in 8 pigs per treatment. ^1^ CTR: control; CTC-ZnO: chlortetracycline + zinc oxide; LY: live yeast; LY-ZnO: live yeast + zinc oxide. Sobs: diversity index; Shannon: Shannon diversity index; Simpson: Simpson diversity index; Ace: Ace diversity index; Chao: bacterial community index.

**Table 8 animals-11-01626-t008:** Effects of live yeast or live yeast combined with zinc oxide on relative abundance of microbiota in feces on d 28 of nursery pigs.

Items (%)	CTR ^1^	CTC-ZnO ^1^	LY ^1^	LY-ZnO ^1^	SEM	*p*-Value
Phylum level						
Firmicutes	58.83 ^ab^	56.11 ^ab^	60.83 ^a^	47.92 ^b^	5.672	0.04
Bacteroidetes	21.17	30.37	32.30	45.58	10.074	0.09
Proteobacteria	15.56	11.31	5.34	4.73	5.164	0.51
Epsilonbacteraeota	2.82	0.46	0.79	0.55	1.123	0.99
Genus level						
*Prevotella_9*	4.73	11.17	13.93	15.94	7.067	0.16
*Succinivibrio*	14.20	10.93	1.67	3.64	8.354	0.24
*Faecalibacterium*	2.86 ^b^	9.22 ^a^	8.49 ^a^	5.69 ^a^	3.163	0.04
*Prevotellaceae_NK3B31_group*	0.31 ^b^	4.86 ^a^	2.31 ^a^	7.53 ^a^	3.460	0.04
*norank_f__Prevotellaceae*	1.63 ^b^	2.55 ^ab^	4.21 ^a^	5.18 ^a^	1.724	0.04
*Anaerovibrio*	0.48 ^b^	4.12 ^a^	1.08 ^a^	3.84 ^a^	1.867	0.03
*Prevotella_2*	0.92	2.32	4.29	1.54	1.603	0.06

Note: SEM is standard error of mean. *n* = 8 indicating the data was analyzed in 8 pigs per treatment. ^a,b^ Within a row, different superscripts mean significant difference (*p* < 0.05). ^1^ CTR: control; CTC: chlortetracycline; LY: live yeast; LY-ZnO: live yeast + zinc oxide.

## Data Availability

All data is contained within the article.

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
