# Peer review of "Live Yeast or Live Yeast Combined with Zinc Oxide Enhanced Growth Performance, Antioxidative Capacity, Immunoglobulins and Gut Health in Nursery Pigs"

_animals, 2021, doi:10.3390/ani11061626_

Round 1
Reviewer 1 Report
Review of the manuscript No 1180183 „Live Yeast or Live Yeast Combined with Zinc Oxide Enhanced Growth Performance, Antioxidative Capacity, Immunoglobulins and Gut Health in Nursery Pigs” for journal ANIMALS
The manuscript is properly prepared. The research problem is not very innovative, as the effectiveness of yeast and zinc oxide in piglet rearing is well known, but it is still important from the point of view of the practical problems of pig production. With regard to the upcoming ban on the use of therapeutic doses of zinc oxide, the group with only yeast seems to be the most important. However the manuscript requires the minor revision.
In the description of the experiment and in the tables, please use the day number of the piglet age (starting from day of life 33, probably), instead of the number of the day of the experiment. Please, provide the piglet body weight in table 2 and in the Result chapter. It is not clear what the "items" mean in table 7 (Sobs, Shannon, Simpson, etc.).
Author Response
Thank you very much for your valuable and helpful comments. Those comments are good for us to improve our paper further. We have studied comments carefully and revised the manuscript thoroughly according to comments, and the amendments are highlighted in red in the revised manuscript. We appreciate for editor and reviewers’ warm work, and hope that the corrections will meet with approval. Once again, thank you for your helpful comments and suggestions. And we look forward to hearing from you about our revised paper.
Kind regards!
Dr. Xiangshu Piao
State Key Laboratory of Animal Nutrition
Ministry of Agriculture and Rural Affairs Feed Industry Centre
China Agricultural University
piaoxsh@cau.edu.cn
2021-05-01
Reviewer 1
Review of the manuscript No 1180183, Live Yeast or Live Yeast Combined with Zinc Oxide Enhanced Growth Performance, Antioxidative Capacity, Immunoglobulins and Gut Health in Nursery Pigs” for journal ANIMALS
Comment: The manuscript is properly prepared. The research problem is not very innovative, as the effectiveness of yeast and zinc oxide in piglet rearing is well known, but it is still important from the point of view of the practical problems of pig production. With regard to the upcoming ban on the use of therapeutic doses of zinc oxide, the group with only yeast seems to be the most important. However, the manuscript requires the minor revision. In the description of the experiment and in the tables, please use the day number of the piglet age (starting from day of life 33, probably), instead of the number of the day of the experiment. Please, provide the piglet body weight in table 2 and in the Result chapter. It is not clear what the "items" mean in table 7 (Sobs, Shannon, Simpson, etc.).
Response: Thanks for the encouragement and helpful suggestion. According to the suggestion, we used the day number of the piglet age in this manuscript: “Weaned on d 32 of the age with 9.2 ± 1.7 kg BW” in abstract and M&M. We have also added the initial BW in Table 2. Besides, the Sobs, Shannon, Simpson, ect. are indexes for measurement of α-diversity in fecal microbiota. We have added the information in Note of Table 7 as following: “Sobs: diversity index; Shannon: Shannon diversity index; Simpson: Simpson diversity index; Ace: Ace diversity index; Chao: bacterial community index.”

Reviewer 2 Report
General comment
The manuscript aimed to investigate the effect of Live yeast and ZnO on performance, antioxidative status, immunoglobulins, and gut health in weaned piglets.
The study is interesting because alternative to in feed antibiotics are needed to reduce antibiotic resistance bacteria.
The manuscript needs few revisions, and some information are needed for better understand and describe the study. More comments are reported below:
Simple Summary
Line 18: Please better specify “weaning stress”.
Abstract
Line 31: Please specify the Phase 1 and phase 2. Did piglets receive treatments?
Line 34: Live yeast, I suggest to include the total colony-forming unit (CFU) supplemented instead. Please also specify if the other treatments were supplemented in both phases, 1 and 2.
Line 36-37 ADG, FCR, specify the acronyms.
Line 42-43: MDA-SOD-CAT, specify the acronyms.
Line 44: are those the two different phases?
Introduction
Line 56: ...is the critical period.... is a critical period.
Line 59: please provide some examples of nutritional strategies, such as Protein content and energy content of the diets. Please also add few sentenced regarding the use of different additives such as natural extract that are largely adopted in swine livestock.
"Natural extracts, have been widely studied for their promising antioxidant, anti-inflammatory and antibacterial properties". Caprarulo, V., Giromini, C., & Rossi, L. (2020). Chestnut and quebracho tannins in pig nutrition: the effects on performance and intestinal health. Animal, 100064.
Line 64-65: Please add few words about the recommendation to the appropriate use of zinc and copper oxide. For example: Many international organizations, such as EFSA and WHO (European Food Safety Authority (EFSA), 2017; World Health Organization (WHO), 2017), have recommended decreasing the use of copper and zinc oxide due to the suspected resistance to certain bacteria and risks associated with the impact of these heavy metals on the environment (Caprarulo et al., 2021)
M&M
Line 138: Please define 500 mg of sample. In M&M section author did not well explained which samples were collected, please specify.
Line 147: in chemical analysis please define all samples.
Line 74: "sIgA " S?
Materials and Methods
Line 97: "Prosaf", Please specify the commercial name of Live Yeast as reported in table 1
Line 101: Please provide SD of weaned day, pleas add this information also in the abstract section.
Line 103: ... trial was contained... better... trial was divided into two experimental phases... then explain the phases adding information about the differences and why the experimental trial was divided.
Line 106-107: Live yeast, I suggest to include the total colony-forming unit (CFU) supplemented instead. Please also specify if the other treatments were supplemented in both phases, 1 and 2.
Line 108: Please better specify if the marker was supplemented for all the experimental treatments.
Line 114: Please specify "feed" did authors mean "Feed refuse"?
Line 116: ADFI/ADG, did authors mean Feed:Gain ratio?
Line 117: I suggest to better explain, briefly, "diarrhea was defined".
Line 121: how authors did collect a representative faecal sample? from the floor? from each piglet in the pen? please provide a detailed description
Line 154: "Prosaf", The commercial name of Live Yeast was not mentioned before, please specify that this product is the live yeast and report this information in the text.
Line 174: Please, in Statistical Analysis section provide information about fixed effect adopted in the model (tretments, time, the interactions?)
Results
I suggest to modify table 2, table 3, table 4, table 5 and table 6 in order to have a clear and easy visualization of the results.
Instead, I suggest to divide the result for each items, such as ADG, ADFI etc... then report the experimental period. I report an explicative example:
ADG
d1 to 7
d 8 to 14
d 15 to d 21
d 22 to d 28
....
Table 7: Please, add in note how authors defined items: Sobs, Shannon, Simpson, Ace, Chao. Fore example:
Chao: bacterial community index.
Shannon: Shannon diversity index.
Discussion
line 271: ...high zinc... better...Pharmacological level, "An increased use of zinc oxide at a pharmacological level (ZnO, 2000–3000 ppm) was observed as an alternative to antibiotics. However, the widespread use of pharmacological levels of ZnO has raised concerns related to environmental issues and the potential increase in the prevalence of antibiotic resistant bacteria.
Caprarulo, V., Hejna, M., Giromini, C., Liu, Y., Dell’Anno, M., Sotira, S., ... & Rossi, L. (2020). Evaluation of Dietary Administration of Chestnut and Quebracho Tannins on Growth, Serum Metabolites and Fecal Parameters of Weaned Piglets. Animals, 10(11), 1945.
Line 272: not also "could pollute the environment ", if ZnO is overused could let to increase antibiotic resistant bacteria.
Line 293-297: This paragraph should be modulated. The authors could Hypnotized that LY could affect intestinal morphology, however this study did not investigate this aspect. Thus, I suggest to do to speculate highlighting this effect on intestinal morphology.
Line 344-346: Please, modulate these sentences in order to be clearer.
Line 347: ...we suspected... I suggest to use another term
Line 348: ...Ly.. should be : LY.
Author Response
Dear editor and reviewers:
Thank you very much for your valuable and helpful comments. Those comments are good for us to improve our paper further. We have studied comments carefully and revised the manuscript thoroughly according to comments, and the amendments are highlighted in red in the revised manuscript. We appreciate for editor and reviewers’ warm work, and hope that the corrections will meet with approval. Once again, thank you for your helpful comments and suggestions. And we look forward to hearing from you about our revised paper.
Kind regards!
Dr. Xiangshu Piao
State Key Laboratory of Animal Nutrition
Ministry of Agriculture and Rural Affairs Feed Industry Centre
China Agricultural University
piaoxsh@cau.edu.cn
2021-05-01
Reviewer 2
Comment: General comment
The manuscript aimed to investigate the effect of Live yeast and ZnO on performance, antioxidative status, immunoglobulins, and gut health in weaned piglets.
The study is interesting because alternative to in feed antibiotics are needed to reduce antibiotic resistance bacteria.
The manuscript needs few revisions, and some information are needed for better understand and describe the study. More comments are reported below:
Response: Thanks for the comments, we have revised the manuscript according to all the comments, please check.
Simple Summary
Comment: Line 18: Please better specify “weaning stress”.
Response: Thanks for the comments. Please refer to Line 18. Weaning stress means “the stress after weaning” which might lead to the increased incidence of diarrhea, body weight loss, and higher mortality.
Comment: Line 31: Please specify the Phase 1 and phase 2. Did piglets receive treatments?
Response: Thanks for the comments, since the abstract was limited to 200 words, we made some revision. Phase 1 is from d 1 to 14, while phase 2 is from d 15 to 28. The piglets received the diets in each treatment.
Comment: Thanks for the comments. Line 34: Live yeast, I suggest to include the total colony-forming unit (CFU) supplemented instead. Please also specify if the other treatments were supplemented in both phases, 1 and 2.
Response: Thanks for the comments. The live yeast in treatments (LY and LY-ZnO) were supplemented in both phases 1 and 2. We added the total colony-forming unit (CFU) information of Saccharomyces cerevisiae (Strain CNCM I-4407, 1010 CFU/g) in Simple summary. Please refer to Line 21.
Comment: Line 36-37 ADG, FCR, specify the acronyms.
Response: Thanks for the comments. We have specified the acronyms in the revised abstract. ADG: Average daily gain; FCR: feed conversion ratio. Please refer to Line 33 and 34 in revised abstract.
Comment: Line 42-43: MDA-SOD-CAT, specify the acronyms.
Response: Thanks for the comments. We have specified the acronyms in the revised abstract. MDA is malondialdehyde, SOD is superoxide dismutase, CAT is catalase. Please refer to Line 33 in the revised abstract.
Comment: Line 44: are those the two different phases?
Response: Thanks for the comments. Since the abstract was limited to 200 words, we made some revision. The experiment mainly contained the phase 1 (d 1 to 14) and phase 2 (d 15 to 28).
Comment:
Introduction
Line 56: ...is the critical period.... is a critical period.
Response: Thanks for the comments. Revised. Weaning is a critical period in commercial pig production systems. Please refer to Line 40.
Comment: Line 59: please provide some examples of nutritional strategies, such as Protein content and energy content of the diets. Please also add few sentenced regarding the use of different additives such as natural extract that are largely adopted in swine livestock.
"Natural extracts, have been widely studied for their promising antioxidant, anti-inflammatory and antibacterial properties". Caprarulo, V., Giromini, C., & Rossi, L. (2020). Chestnut and quebracho tannins in pig nutrition: the effects on performance and intestinal health. Animal, 100064.
Response: Thanks for the comments. We have added an few sentenced regarding the use of different additives such as natural extract that are largely adopted in swine livestock as following: The use of different additives such as natural extracts are largely adopted in swine livestock [2], which have been widely studied for their promising antioxidant, anti-inflammatory and antibacterial properties [3]. Please refer to Line 44-47.
Comment: Line 64-65: Please add few words about the recommendation to the appropriate use of zinc and copper oxide. For example: Many international organizations, such as EFSA and WHO (European Food Safety Authority (EFSA), 2017; World Health Organization (WHO), 2017), have recommended decreasing the use of copper and zinc oxide due to the suspected resistance to certain bacteria and risks associated with the impact of these heavy metals on the environment (Caprarulo et al., 2021).
Response: Thanks for the comments. We added few words about the recommendation to the appropriate use of zinc recommended as following: Many international organizations, such as European Food Safety Authority and World Health Organization, have recommended decreasing the use of copper and ZnO due to the suspected resistance to certain bacteria and risks associated with the impact of these heavy metals on the environment [2]. Please refer to Line 53-57.
Comment: M&M
Line 138: Please define 500 mg of sample. In M&M section author did not well explained which samples were collected, please specify.
Response: Thanks for the comments. We have revised the M&M section and better explained the samples collection in M&M section. Please refer to Line 123, 137 and 148.
Comment: Line 147: in chemical analysis please define all samples.
Response: Thanks for the comments. The chemical analysis was done for in the dried fecal samples and feed. We defined all samples in chemical analysis section as following: The samples for ATTD of nutrients were the dried fecal samples and feed. Please refer to Line 129 and 132.
Comment: Line 74: "sIgA " S?
Response: Thanks for the comments. sIgA has been changed into “SIgA”. Please refer to Line 65.
Comment: Materials and Methods
Line 97: "Prosaf", Please specify the commercial name of Live Yeast as reported in table 1
Response: Thanks for the comments. We have added the information for the commercial product “Prosaf” in the note in table 1. Prosaf was provided by Phileo Lesaffre Animal Care. Please refer to Line 106-108. The product "Prosaf" is a yeast extract obtained by autolysis of proprietary Saccharomyces cerevisiae baker’s yeast strains. The product is a sustainable source of highly digestible and palatable protein and peptides rich in essential amino acids and glutamate.
Comment: Line 101: Please provide SD of weaned day, please add this information also in the abstract section.
Response: Thanks for the comments. All the piglets were weaned on d 32 of the age with 9.2 ± 1.7 kg BW. Please refer to Line 92.
Comment: Line 103: ... trial was contained... better... trial was divided into two experimental phases... then explain the phases adding information about the differences and why the experimental trial was divided.
Response: Thanks for the comments. The trial was divided into two experimental phases: phase 1 (d 1 to 14) and 2 (d 15 to 28). Phase 1 is the first two weeks after weaning, it is the phase that weaning stress usual happens. Phase 2 is 3-4 week after weaning is the second two weeks after weaning to investigate the accumulated and long-term effect of the feed additives.
Comment: Line 106-107: Live yeast, I suggest to include the total colony-forming unit (CFU) supplemented instead. Please also specify if the other treatments were supplemented in both phases, 1 and 2.
Response: Thanks for the comments. We have described this information in 2.1. Experimental Products. Please refer to Line 88-89. The live yeast in treatments (LY and LY-ZnO) were supplemented in both phases 1 and 2. All the treatments were supplemented in both phases 1 and 2. Please refer to Line 99.
Comment: Line 108: Please better specify if the marker was supplemented for all the experimental treatments.
Response: Thanks for the comments. Chromic oxide (2.5 g/kg at the end of trail to detect the nutrient retention) was used as an indigestible marker, which was supplemented for all the experimental treatments. Please refer to Line 102
Comment: Line 114: Please specify "feed" did authors mean "Feed refuse"?
Response: Thanks for the comments. Feed means diet.
Comment: Line 116: ADFI/ADG, did authors mean Feed:Gain ratio?
Response: Thanks for the comments. Yes, Feed:Gain ratio is FCR (ADFI/ADG).
Comment: Line 117: I suggest to better explain, briefly, "diarrhea was defined".
Response: Thanks for the comments. We use diarrhea score to measure the diarrhea rate as following: Diarrhea score was recorded daily, diarrhea was defined and diarrhea rate was calculated as following: diarrhea rate (%) = (the diarrhea days × the number of diarrhea pigs)/(the total experiment days × the total number of pigs) × 100 [21]. Please refer to Line 118-121.
Comment: Line 121: how authors did collect a representative faecal sample? from the floor? from each piglet in the pen? please provide a detailed description
Response: Thanks for the comments. We collect the representative faecal sample from each piglet in the pen, please refer to Line 123-124. The description is as following: “In each pen, about 1 kg fresh fecal samples from each piglets in the pen were collected using the grab sample technique according to Long et al. [21] from d 12 to 14 and d 26 to 28.”
Comment: Line 154: "Prosaf", The commercial name of Live Yeast was not mentioned before, please specify that this product is the live yeast and report this information in the text.
Response: Thanks for the comments. We have added the information of "Prosaf" in the note of Table 1. Please refer to Line 106-108. The product "Prosaf" is a yeast extract obtained by autolysis of proprietary Saccharomyces cerevisiae baker’s yeast strains. The product is a sustainable source of highly digestible and palatable protein and peptides rich in essential amino acids and glutamate,
Comment: Line 174: Please, in Statistical Analysis section provide information about fixed effect adopted in the model (treatments, time, the interactions?)
Response: Thanks for the comments. Please refer to Line 173-175. We have added information about fixed effect adopted in the model as following: A randomized complete block design was used in this study with sex and initial BW as blocking criteria. Mixed procedure of SAS (version 9.2, 2008) [26] was used for statistical analysis. Dietary treatment was the fixed effect, sex and initial BW were the random effects.
Comment: Results
I suggest to modify table 2, table 3, table 4, table 5 and table 6 in order to have a clear and easy visualization of the results.
Instead, I suggest to divide the result for each items, such as ADG, ADFI etc... then report the experimental period. I report an explicative example:
ADG
d1 to 7
d 8 to 14
d 15 to d 21
d 22 to d 28
Response: Thanks for the comments. We have revised Table 2 according to the advice. Please refer to Table 2.
Comment: Table 7: Please, add in note how authors defined items: Sobs, Shannon, Simpson, Ace, Chao. For example:
Chao: bacterial community index.
Shannon: Shannon diversity index.
Response: Thanks for the comments. We have added theinformation in the note of Table 7 (Line 253-254): Sobs: diversity index; Shannon: Shannon diversity index; Simpson: Simpson diversity index; Ace: Ace diversity index; Chao: bacterial community index.
Comment: Discussion
line 271: ...high zinc... better...Pharmacological level, "An increased use of zinc oxide at a pharmacological level (ZnO, 2000–3000 ppm) was observed as an alternative to antibiotics. However, the widespread use of pharmacological levels of ZnO has raised concerns related to environmental issues and the potential increase in the prevalence of antibiotic resistant bacteria.
Caprarulo, V., Hejna, M., Giromini, C., Liu, Y., Dell’Anno, M., Sotira, S., ... & Rossi, L. (2020). Evaluation of Dietary Administration of Chestnut and Quebracho Tannins on Growth, Serum Metabolites and Fecal Parameters of Weaned Piglets. Animals, 10(11), 1945.
Response: Thanks for the comments. We have added this sentence in the discussion as following (Line 263-270): High dose of inorganic Zn (2,000–4,000 mg/kg of Zn as ZnO) has been widely used in the diets of piglets in the swine industry, due to its effects on increasing growth performance and alleviating post-weaning diarrhea [27]. An increased use of zinc oxide at a pharmacological level (ZnO, 2,000-3,000 mg/kg) was also observed as an alternative to antibiotics. However, the widespread use of pharmacological levels of ZnO has raised concerns related to environmental issues and the potential increase in the prevalence of antibiotic resistant bacteria [2].
Comment: Line 272: not also "could pollute the environment ", if ZnO is overused could led to increase antibiotic resistant bacteria.
Response: Thanks for the comments. Please refer to Line 270-271. We corrected this sentence into “However, the overuse of ZnO could pollute the environment and could led to increase antibiotic resistant bacteria since the zinc could not to be fully utilized by pigs.” according to the comments.
Comment: Line 293-297: This paragraph should be modulated. The authors could Hypnotized that LY could affect intestinal morphology, however this study did not investigate this aspect. Thus, I suggest to do to speculate highlighting this effect on intestinal morphology.
Response: Thanks for the comments. We have deleted this this sentence about the effect on intestinal morphology since this study did not investigate this aspect
Comment: Line 344-346: Please, modulate these sentences in order to be clearer.
Response: Thanks for the comments, please refer to Line 340-341, we have modulated these sentences according to the advice as following: In order to figure out the possible mechanism of LY on modulating gut health in nursery pigs, we investigated the fecal microbiota community and fecal VFA contents.
Comment: Line 347: ...we suspected... I suggest to use another term
Response: Thanks for the comments. We have deleted “we suspected”. Please refer to Line 362.
Comment: Line 348: ...Ly.. should be : LY.
Response: Thanks for the comments. Revised. Please refer to Line 363.

Reviewer 3 Report
The manuscript describes the effect of dietary live yeast (LY), alone or in combination with Zn0, on growth performance (including diarrea rate), ATTD of nutrients, serum antioxidant enzymes and immunoglobulins concentrations, fecal contents (or profile) of VFA and microbiota in weanling piglets. The period studied covered the first 28 post-weaning days with experimental design involving 4 diets: the 2 experimental diets with LY mentioned above, and 2 control diets: a negative dietary treatment (basal diet) and positive dietary treatment (antibiotic+Zn0).
My first impression is rather poor as it is difficult to read. The reasons are mainly the English writing style, as well as the non-uniform structure of the text. For this reason, I would recommend them that the paper is fully reviewed by a native speaker (or better by a English editing service). But prior to English editing, they have to think about providing the same structure and order throughout the text or different sections (including tables). For example, and taking into account the current draft:
The order has been established according to the independent text paragraphs, or numbered sub-sections
Variables studied |
Material and Methods |
Results |
Discussion |
Tables1 |
Growth performance |
1A |
1 |
1 |
1 |
Diarrhea rate |
1B |
2 |
5 |
1 |
ATTD of nutrients |
2 |
5 |
2 |
4 |
Serum IGs |
3A |
4 |
3 |
3 |
Serum AOX enzymes |
3B |
3 |
4 |
2 |
Fecal VFA contents |
4 |
6 |
7 |
5 |
Fecal microbiota |
5 |
7 |
6 |
6 and 7 |
1 The Table 1 corresponding to chemical composition of basal diet has not taken into account, thus the number is the number of each table minus 1.
It is obvious that the presentation is chaotic, unacceptable to be submitted to a high impact scientific journal. Also, the presentation of results within sub-section could be improved (e.g., 3.1 Performance).
In addition to formal and presentation issues, there are other substantive concerns that need to be addressed. For example, I’m going to expose my doubts in relation to the statistical analysis and sample size in some variables.
Authors wrote: “Mixed procedure of SAS was used for statistical analyses. The difference of diarrhea rate was analyzed by chi-square contingency test, while the statistical differences of other data except for the microbiota were analyzed via the Student- Neuman-Keul’s Multiple Range Tests (L175-178)…. Differences of microbiota abundance in feces were analyzed using Kruskal-Wallis 180 rank sum test (L180-181)”.
A mixed linear model was used; however, this model is not properly described. What were the fixed and random factors? It seems that dietary treatment was the fixed factor, but which was the random effect? For which variables was it used? I assume those that were normally distributed, such as growth performance and digestibility, but these are not detailed. On the other hand, it also seems that non-parametric tests were performed for other data (non-normally distributed), such serum and fecal variables. However, the non-parametric test was different according to the parameter. Why? Which variables were truly normally distributed or not? Finally, authors can also see that the sentence about the experimental units considered is in the middle of the description about analyses or tests applied (L178-180).
In relation to the last sentence, most of variables (except for performance and diarrhea rate) were analysed taking into account individual data at week (Table 3, Table 4, ) or biweekly intervals (Table 5, Table 6), or even not detailed[i] (Table 7 and 8). The sample size was n=8; however, it is not clear if were 8 animals per treatment or in total. How were these animals sampled? One animal per pen and treatment, or they were sampled without taking in mind the pen or sex? It is not also detailed. On the other hand, I supposed that fecal samples were pooled samples not individual, although the sample size was 8 (one per pen).
Finally, the sample size seems limited. In this sense, I wonder if it would not have been better to increase the sample size at the end of the study rather than increase the number of sampling times; mainly taking into account that a repeated measures model was not considered.
I hope that this revision will give the authors food for thought, and then re-write and re-analyse the manuscript. It should be fully checked as there are countless minor corrections (abstract of less than 200 words; explicit extended acronyms; detail model, manufacturer, city, and country in all products or devices; remove the year in the citations in the text; units and format units; ingredients by proportion in Table 1; SEM with more precision than means (one decimal place more); avoid citations from technical papers such as number 25,…)
[i] According to L144, fecal samples for them were taken on day 28, but also for Table 6.
Author Response
Dear editor and reviewers:
Thank you very much for your valuable and helpful comments. Those comments are good for us to improve our paper further. We have studied comments carefully and revised the manuscript thoroughly according to comments, and the amendments are highlighted in red in the revised manuscript. We appreciate for editor and reviewers’ warm work, and hope that the corrections will meet with approval. Once again, thank you for your helpful comments and suggestions. And we look forward to hearing from you about our revised paper.
Kind regards!
Dr. Xiangshu Piao
State Key Laboratory of Animal Nutrition
Ministry of Agriculture and Rural Affairs Feed Industry Centre
China Agricultural University
piaoxsh@cau.edu.cn
2021-05-01
Comment:
Review 3
The manuscript describes the effect of dietary live yeast (LY), alone or in combination with ZnO, on growth performance (including diarrhea rate), ATTD of nutrients, serum antioxidant enzymes and immunoglobulins concentrations, fecal contents (or profile) of VFA and microbiota in weanling piglets. The period studied covered the first 28 post-weaning days with experimental design involving 4 diets: the 2 experimental diets with LY mentioned above, and 2 control diets: a negative dietary treatment (basal diet) and positive dietary treatment (antibiotic + ZnO).
My first impression is rather poor as it is difficult to read. The reasons are mainly the English writing style, as well as the non-uniform structure of the text. For this reason, I would recommend them that the paper is fully reviewed by a native speaker (or better by a English editing service). But prior to English editing, they have to think about providing the same structure and order throughout the text or different sections (including tables). For example, and taking into account the current draft:
The order has been established according to the independent text paragraphs, or numbered sub-sections
Variables studied |
Material and Methods |
Results |
Discussion |
Tables1 |
Growth performance |
1A |
1 |
1 |
1 |
Diarrhea rate |
1B |
2 |
5 |
1 |
ATTD of nutrients |
2 |
5 |
2 |
4 |
Serum IGs |
3A |
4 |
3 |
3 |
Serum AOX enzymes |
3B |
3 |
4 |
2 |
Fecal VFA contents |
4 |
6 |
7 |
5 |
Fecal microbiota |
5 |
7 |
6 |
6 and 7 |
1 The Table 1 corresponding to chemical composition of basal diet has not taken into account, thus the number is the number of each table minus 1.
It is obvious that the presentation is chaotic, unacceptable to be submitted to a high impact scientific journal. Also, the presentation of results within sub-section could be improved (e.g., 3.1 Performance).
Response: Thanks for the comments. We have invited a native English teacher to help us revise the manuscript to make it easier for reading. And we revised the structure of this manuscript as following in this manuscript to make the structure better for reading, we also improve the presentation of results within sub-section. Please check as following:
Variables studied |
Material and Methods |
Results |
Discussion |
Tables1 |
Growth performance |
1A |
1A |
1A |
2 |
Diarrhea rate |
1B |
1B |
1B |
2 |
ATTD of nutrients |
2 |
2 |
2 |
3 |
Serum IGs |
3 |
3 |
3 |
4 |
Serum AOX enzymes |
4 |
4 |
4 |
5 |
Fecal VFA contents |
5 |
5 |
5 |
6 |
Fecal microbiota |
6 |
6 |
6 |
7 and 8 |
Comment: In addition to formal and presentation issues, there are other substantive concerns that need to be addressed. For example, I’m going to expose my doubts in relation to the statistical analysis and sample size in some variables.
Authors wrote: “Mixed procedure of SAS was used for statistical analyses. The difference of diarrhea rate was analyzed by chi-square contingency test, while the statistical differences of other data except for the microbiota were analyzed via the Student-Neuman-Keul’s Multiple Range Tests (L175-178)….Differences of microbiota abundance in feces were analyzed using Kruskal-Wallis rank sum test (L180-181)”.
A mixed linear model was used; however, this model is not properly described. What were the fixed and random factors? It seems that dietary treatment was the fixed factor, but which was the random effect? For which variables was it used? I assume those that were normally distributed, such as growth performance and digestibility, but these are not detailed. On the other hand, it also seems that non-parametric tests were performed for other data (non-normally distributed), such serum and fecal variables. However, the non-parametric test was different according to the parameter. Why? Which variables were truly normally distributed or not? Finally, authors can also see that the sentence about the experimental units considered is in the middle of the description about analyses or tests applied (L178-180).
Response: Thanks for the comments. In this study, we used a mixed model. We have corrected and made the description more suitable for reading. Please refer to Line 173-181.
In this study, a randomized complete block design was used in this study with sex and initial BW as blocking criteria. Dietary treatment was fixed effect, sex and initial BW were random effects.
Moreover,in this study, the non-parametric test was used in the analysis of diarrhea rate and microbiota community data, while the parametric test was used in other data (performance, nutrient digestibility, serum parameters (Antioxidant status and immunoglobulins).
We have revised the Statistical Analysis as following: “A randomized complete block design was used in this study with sex and initial BW as blocking criteria. Mixed procedure of SAS (version 9.2, 2008) [26] was used for statistical analysis. Dietary treatment was fixed effect, sex and initial BW were random effects. For performance and diarrhea rate, individual pen was used as statistical unit, while for other data, the individual pig was used as statistical unit. The difference of diarrhea rate was analyzed by chi-square contingency test, while the statistical differences of other data except for the microbiota were analyzed via the Student-Neuman-Keul’s Multiple Range Tests. Differences of microbiota abundance in feces were analyzed using Kruskal-Wallis rank sum test. Significant difference was defined as p ≤ 0.05, and a trend of difference was determined as 0.05 < p ≤ 0.10.”
Comment: In relation to the last sentence, most of variables (except for performance and diarrhea rate) were analysed taking into account individual data at week (Table 3, Table 4,) or biweekly intervals (Table 5, Table 6), or even not detailed[i] (Table 7 and 8). The sample size was n=8; however, it is not clear if were 8 animals per treatment or in total. How were these animals sampled? One animal per pen and treatment, or they were sampled without taking in mind the pen or sex? It is not also detailed. On the other hand, I supposed that fecal samples were pooled samples not individual, although the sample size was 8 (one per pen).
Response: Thanks for the comments, we have detailed the time for VFA and fecal microbiota community (on d 28) and corrected according to the comments.
We have added the information about how we collect samples in M&M (one barrow weighing near the average BW in each pen).
According to the references, n = 8 means we collected 8 samples in each replicate pen of each treatment for the sample analysis (Line 137 and 148). We hope to add more samples in the future studies.
Comment: Finally, the sample size seems limited. In this sense, I wonder if it would not have been better to increase the sample size at the end of the study rather than increase the number of sampling times; mainly taking into account that a repeated measures model was not considered.
Response: Thanks for the comments and suggestions. We will try to increase the sample size at the end of the study rather than increase the number of sampling times in the future study. In this study, we analyzed performance and samples on d 1, 7, 14 and 28 (Every week) to determine if LY or LY combined with ZnO could improve performance and health of nursery pigs week by week and try to explain the possible mechanism.
Although 8 samples (n = 8) as standards for each treatment is a bit limited, we think it could still be used for the data analysis. This number of samples would be sufficient to detect the serum parameters, VFA and microbiota community according to previous studies (such as Li et al., 2016. Lactobacillus acidophilus alleviates the inflammatory response to enterotoxigenic Escherichia coli K88 via inhibition of the NF-kappaB and p38 mitogen-activated protein kinase signaling pathways in piglets. BMC Microbiol. 16: 273; Hu et al., 2019. Effects of early-life lactoferrin intervention on growth performance, small intestinal function and gut microbiota in suckling piglets. Food & Function, 2019, 10: 5361-5373; Kibeom Jang and Sung Woo Kim. Supplemental effects of dietary nucleotides on intestinal health and growth performance of newly weaned pigs. Journal of Animal Science, 2019, 4875–4882).
For the next animal growth performance trial, we would have more replicates to more accurately detect the difference. Thanks again for the helpful comments.
Comment: I hope that this revision will give the authors food for thought, and then re-write and re-analyse the manuscript. It should be fully checked as there are countless minor corrections (abstract of less than 200 words; explicit extended acronyms; detail model, manufacturer, city, and country in all products or devices; remove the year in the citations in the text; units and format units; ingredients by proportion in Table 1; SEM with more precision than means (one decimal place more); avoid citations from technical papers such as number 25,…)
Response: Thanks for the comments. We have made minor correction of all the mistakes in this manuscript. In detail, we revised the abstract to make it less than 200 words (Line 26-38).
We explicit extended acronyms and added the detail model, manufacturer, city, and country in all products or devices (Line 139, 142; Line 173-175). We remove the year in the citations in the text (Such as Line 103, 124, 152).
Besides, we rechecked all the units and format units and the ingredients by proportion in Table 1 (Please refer to Table 1), and we rechecked the SEM and avoided citations from technical papers such as number 25. We use another reference instead of the previous number 25 (Line 468-470).
Comment: [i] According to L144, fecal samples for them were taken on day 28, but also for Table 6.
Response: Thanks for the comments. Revised. We added this information in the title of Table 6.

Round 2
Reviewer 2 Report
The authors largely addressed the review comments and the text has been extensively revised following previous comments.
In accordance with my previous comments, I suggest to modify table 3, table 4, table 5 and table 6 in order to have a clear and easy visualization of the results as authors did for the table 2. This will make tables more readable and easier to understand. I report below the previous comment:
“I suggest to modify table 2, table 3, table 4, table 5 and table 6 in order to have a clear and easy visualization of the results.
Instead, I suggest to divide the result for each items, such as ADG, ADFI etc... then report the experimental period.”
Author Response
Thank you very much for your valuable and helpful comments. Those comments are good for us to improve our paper further. We have studied comments carefully and revised the manuscript thoroughly according to comments, and the amendments are highlighted in red in the revised manuscript. We appreciate for editor and reviewers’ warm work, and hope that the corrections will meet with approval. Once again, thank you for your helpful comments and suggestions. And we look forward to hearing from you about our revised paper.
Kind regards!
Dr. Xiangshu Piao
State Key Laboratory of Animal Nutrition
Ministry of Agriculture and Rural Affairs Feed Industry Centre
China Agricultural University
piaoxsh@cau.edu.cn
2021-05-17
Reviewer 2:
Comment: The authors largely addressed the review comments and the text has been extensively revised following previous comments.
In accordance with my previous comments, I suggest to modify table 3, table 4, table 5 and table 6 in order to have a clear and easy visualization of the results as authors did for the table 2. This will make tables more readable and easier to understand. I report below the previous comment:
“I suggest to modify table 2, table 3, table 4, table 5 and table 6 in order to have a clear and easy visualization of the results.
Instead, I suggest to divide the result for each items, such as ADG, ADFI etc... then report the experimental period.”
Response: Thanks for the useful suggestion. We have modified table 2, table 3, table 4, table 5 and table 6 in order to have a clear and easy visualization of the results in the revised version. Please refer to the revised table 2, table 3, table 4, table 5 and table 6.

Reviewer 3 Report
After the first round of revision, the manuscript has obviously improved in structure and order in the text, but nevertheless I still have some major concerns regarding the statistical analysis (A) and the presentation of results (B).
A. Statistical analysis
Authors wrote (L173-174): “Dietary treatment was the fixed effect, sex and initial BW were the random effects. For performance and diarrhea rate, individual pen was used as statistical unit, while for other data, the individual pig was used as statistical unit.” Previously, they also wrote (L94): “6 piglets (3 barrows and 3 gilts) per pen” and (L135-136): “barrows weighing near the average BW in each pen (n = 8) was used for the collection of blood samples”
The model used does not seem to make sense. Are the authors sure that they have used a mixed model? A mixed procedure of SAS does not necessarily imply a mixed model. For example, how and why the sex effect is considered as a random factor? For growth performance data, there were 8 pens per treatment. What level of sex effect was assigned to each pen if they were balanced (50:50)? For individual blood data, how was it possible to consider the sex effect as random if there was only one level (only barrows)?
Are you sure that the initial BW was considered as a random effect? or as a covariate? For which variables was the initial BW considered? For all of them? Does it make sense to consider it as a random effect?
THE AUTHORS SHOULD DETAIL EXACTLY THE ANALYTIC MODEL USED (EQUATION) FOR EACH DATA SET IN THEIR NEXT REPLY. IT IS NOT CLEAR, AND IS A KEY POINT IN ACCEPTING/REJECTING THE PAPER.
B. Results
- All significant results reported in the tables should be described in the text. For example, there was significant effect on catalase activity (Table 5; p = 0.02), which is omitted in the text.
- Authors should either describe all the trends found (p < 0.1) or not comment on any of them. For example, see some tendencies described in L224-225 and 228-230, while others p-values < 0.1 are omitted (i.e. Table 7 and 8).
On the other hand, Authors wrote (L211-212 and 213-214): On d 7, 14, 21 and 28, the serum SOD content in pigs fed LY-ZnO was increased (p < 0.05) compared with CTR… On d 7 and 21, the serum SOD content in pigs fed CTC-ZnO was increased (p < 0.05) compared with CTR. The p-value for SOD content on day 21 reported in Table 5 was not significant (p = 0.38). Please, check all the results section carefully.
Please, provide sample size (n) in each table indicating the kind of data analyzed (pen or animal).
List the ingredients by proportion in Table 1. For example, list soy oil before fish meal, or limestone before dicalcium phospate.
Provide the SEM with more precision than means (one decimal place more) in all tables.
In addition, there are other corrections and suggestions through the paper:
Please, replace SIgA by secretory IgA or simply by IgA(L65)
From Phileo by Lesaffre (CITY, France) (L88-89) Which city? Marquette-lez-Lille?
Chromic oxide (2.5 g/kg at the end of trail to detect the nutrient retention) was used as an indigestible marker (L99-199) Chromic oxide was added at the end of the trial or was it added as an ingredient to the basal diets as reported in Table 2? Please, delete “at the end of trail to detect the nutrient retention” to avoid any misunderstanding.
From my point of view, from the middle of line 114 to line 119 could be added to the subsection 2.3, with a new heading such as “Data Recording, Sample Collection and Analysis”. It would seem more logical to group the data to be analyzed.
L116-117: Diarrhea score was recorded daily, diarrhea was defined and diarrhea rate was calculated. What does it mean? How was it defined?
Please, check “fresh fecal samples from each piglets in the pen” (L121) From each piglet? Was it just one piglet? From each pen? How many piglets per pen?
About text from L123-132 and variables in Table 3: Authors should think about the most logical order in which to arrange the variables, and then describe the analyses in the same order. For example, organic matter should be mentioned before crude protein and ether extract, as these fractions are included in the former. Dry matter is described earlier in the text, but appears later in the table… Which is Gr (L126)? Chromium (Cr)? This one would be analyzed to determine the digestibility coefficients (L132-134), however other analyses, such as fiber and energy, were described previously. Please, check it.
L132: Please, replace (Parr 1281, Automatic Energy Analyzer; Moline, IL) by (Parr 1281 Automatic Energy Analyzer, Moline, IL, USA)
L140: Please, replace , Texa, USA by Montgomery, TX, USA
Please, check the order of antioxidant variables in the text is similar and in accordance with the presentation of the data in the table 5.
Avoid starting sentences with an acronymus or similar (i.e., L148, L157,…)
L149: Please, replace Pennsylvania by Avondale, PA, USA
L151: Used the gas chromatograph sample bottle to transfer the supernatant. Please, review English editing.
L157: Please, replace nanodrop2000 by NanoDrop 2000 Spectrophotometer, and adding the manufacturer
L160: Please replace axygen Biosciences, CA, USA by Axygen Biosciences, Union City, CA, USA
L 161: Please replace Waltham, USA by Waltham, MA, USA
L163: Please replace San Diego, USA by San Diego, CA,USA
L165: Trimmatic or Trimmomatic???. Please, check it. Authors described the version used of this software, but no others such as flash software or upars software. Please, be consistent.
L155-169: On the other hand, I’m not sure so I’m not an expertise in sequence analyses if some acronymus and/or units should be extended or not in this paragraph. For example, BP (base pair), RDP (Ribosomal Database Project), and QIIME (Quantitative Insights Into Microbial Ecology).
Change the references to Table 6 (L237) and Table 7 (L241) by 7 and 8, respectively.
Finally, and taking into account that the references are numbered consecutively, it may not be necessary to differentiate publications of the same author and year with a and b. For example, references number 2 and 3, or 12 and 43. Please, check it according to the journal guidelines.
Author Response
Thank you very much for your valuable and helpful comments. Those comments are good for us to improve our paper further. We have studied comments carefully and revised the manuscript thoroughly according to comments, and the amendments are highlighted in red in the revised manuscript. We appreciate for editor and reviewers’ warm work, and hope that the corrections will meet with approval. Once again, thank you for your helpful comments and suggestions. And we look forward to hearing from you about our revised paper.
Kind regards!
Dr. Xiangshu Piao
State Key Laboratory of Animal Nutrition
Ministry of Agriculture and Rural Affairs Feed Industry Centre
China Agricultural University
piaoxsh@cau.edu.cn
2021-05-17
Reviewer 3:
Comment: After the first round of revision, the manuscript has obviously improved in structure and order in the text, but nevertheless I still have some major concerns regarding the statistical analysis (A) and the presentation of results (B).
- Statistical analysis
Authors wrote (L173-174): “Dietary treatment was the fixed effect, sex and initial BW were the random effects. For performance and diarrhea rate, individual pen was used as statistical unit, while for other data, the individual pig was used as statistical unit.” Previously, they also wrote (L94): “6 piglets (3 barrows and 3 gilts) per pen” and (L135-136): “barrows weighing near the average BW in each pen (n = 8) was used for the collection of blood samples”
The model used does not seem to make sense. Are the authors sure that they have used a mixed model? A mixed procedure of SAS does not necessarily imply a mixed model. For example, how and why the sex effect is considered as a random factor? For growth performance data, there were 8 pens per treatment. What level of sex effect was assigned to each pen if they were balanced (50:50)? For individual blood data, how was it possible to consider the sex effect as random if there was only one level (only barrows)? Are you sure that the initial BW was considered as a random effect? or as a covariate? For which variables was the initial BW considered? For all of them? Does it make sense to consider it as a random effect?
THE AUTHORS SHOULD DETAIL EXACTLY THE ANALYTIC MODEL USED (EQUATION) FOR EACH DATA SET IN THEIR NEXT REPLY. IT IS NOT CLEAR, AND IS A KEY POINT IN ACCEPTING/REJECTING THE PAPER.
Response: Thanks for the useful and kindful suggestion. We were fully sorry for the mistakes about random effect description, and we have carefully rechecked and revised the data analysis part in this manuscript according to the comments. In this study, we used a mixed model, and the sex and initial BW were not considered as random effects, while the block was considered as the random effect. A randomized complete block design was used in this study with sex and initial BW as blocking criteria. For the growth performance, diarrhea rate and nutrient digestibility data, there were 8 pens (as 8 replicates) per treatment. For the blood and fecal samples, there are 8 pigs (as 8 replicates) per treatment.
The data analysis part (Line 177-186) has been changed into: A randomized complete block design was used in this study with sex and initial BW as blocking criteria. Mixed procedure of SAS (version 9.2, 2008) [26] was used for statistical analysis. Dietary treatment was the fixed effect, while the block was the random effect. For performance, diarrhea rate and nutrient digestibility, individual pen was used as statistical unit, while for other data, the individual pig was used as statistical unit. The difference of diarrhea rate was analyzed by chi-square contingency test, while the statistical differences of other data except for the microbiota were analyzed via the Student-Neuman-Keul’s Multiple Range Tests. Differences of microbiota abundance in feces were analyzed using Kruskal-Wallis rank sum test. Significant difference was defined as p ≤ 0.05, and a trend of difference was determined as 0.05 < p ≤ 0.10.
Comment: B. Results
- All significant results reported in the tables should be described in the text. For example, there was significant effect on catalase activity (Table 5; p = 0.02), which is omitted in the text.
Response: Thanks for the suggestion, we have added significant effect on catalase activity (Table 5; p = 0.02) in the result part. Please refer to Line 226-228: On d 21, the serum CAT level in pigs fed CTR was lower (p < 0.05) compared with other treatments (Table 5).
Comment: - Authors should either describe all the trends found (p < 0.1) or not comment on any of them. For example, see some tendencies described in L224-225 and 228-230, while others p-values < 0.1 are omitted (i.e. Table 7 and 8).
Response: Thanks for the suggestion, we have deleted all the description about the trend found (p < 0.1). For example: we deleted “Pigs fed LY-ZnO were tended to have increased concentrations of isovaleric acid (p = 0.08) and valeric acid (p = 0.06) compared with CTR and Pigs fed LY-ZnO tended to have increased concentrations of acetic acid (p = 0.07) and valeric acid (p = 0.09) compared with CTR in Table 7.”
Comment: On the other hand, Authors wrote (L211-212 and 213-214): On d 7, 14, 21 and 28, the serum SOD content in pigs fed LY-ZnO was increased (p < 0.05) compared with CTR… On d 7 and 21, the serum SOD content in pigs fed CTC-ZnO was increased (p < 0.05) compared with CTR. The p-value for SOD content on day 21 reported in Table 5 was not significant (p = 0.38). Please, check all the results section carefully.
Response: Thanks for the suggestion, we are fully sorry for the mistake about data of SOD on d 21 in serum. We have carefully rechecked, revised the data and description about the result part in the manuscript. Please refer to the revised Table 5.
Comment: Please, provide sample size (n) in each table indicating the kind of data analyzed (pen or animal).
Response: Thanks for the suggestion, we have added note about sample size (n) in each table indicating the kind of data analyzed (pen or animal) in every table. In detail, we added “N = 8 indicating the data was analyzed in 8 pens per treatment” in the note for the performance, diarrhea rate and nutrient digestibility data, while we added “N = 8 indicating the data was analyzed in 8 pigs per treatment” in the note for the blood and fecal VFA and microbiota data.
Comment: List the ingredients by proportion in Table 1. For example, list soy oil before fish meal, or limestone before dicalcium phospate.
Response: Thanks for the suggestion, we have revised and listed the ingredients by proportion in the revised Table 1.
Comment: Provide the SEM with more precision than means (one decimal place more) in all tables.
Response: Thanks for the suggestion, we have provided the SEM with more precision than means (one decimal place more) in all tables. Please refer to all the revised Tables.
Comment: In addition, there are other corrections and suggestions through the paper: Please, replace SIgA by secretory IgA or simply by IgA(L65)
Response: Thanks for the suggestion. We have revised according to the comment, please refer to Line 65.
Comment: From Phileo by Lesaffre (CITY, France) (L88-89) Which city? Marquette-lez-Lille?
Response: Thanks for the suggestion. We have added the city for Phileo by Lesaffre which is Marcq-en-Baroeul in France. Please refer to Line 89.
Comment: Chromic oxide (2.5 g/kg at the end of trail to detect the nutrient retention) was used as an indigestible marker (L99-199) Chromic oxide was added at the end of the trial or was it added as an ingredient to the basal diets as reported in Table 2? Please, delete “at the end of trail to detect the nutrient retention” to avoid any misunderstanding.
Response: Thanks for the suggestion. We have revised according to the comment, the chromic oxide was added as an ingredient to the basal diets as reported in Table 2. We have deleted “at the end of trail to detect the nutrient retention” to avoid any misunderstanding. Please refer to Line 100-103.
Comment: From my point of view, from the middle of line 114 to line 119 could be added to the subsection 2.3, with a new heading such as “Data Recording, Sample Collection and Analysis”. It would seem more logical to group the data to be analyzed.
Response: Thanks for the suggestion. We have revised according to the comment, please refer to Line 113-124.
Comment: L116-117: Diarrhea score was recorded daily, diarrhea was defined and diarrhea rate was calculated. What does it mean? How was it defined?
Response: Thanks for the suggestion. We have added the information about Diarrhea score and diarrhea rate in Line 119-123: A scoring system was applied to indicate the presence and severity of diarrhea as follows: 1 = hard feces; 2 = slightly soft feces; 3 = soft, partially formed feces; 4 = loose, semiliquid feces; and 5 = watery, mucous-like feces. When the average score was over 3, pigs were identified as having diarrhea.
Comment: Please, check “fresh fecal samples from each piglet in the pen” (L121) From each piglet? Was it just one piglet? From each pen? How many piglets per pen?
Response: Thanks for the reminding, the fecal samples from each pen was collected for the ATTD of nutrient. Each collection of feces for 3 d was pooled by pen and dried at 65 °C for 72 h. The feces were from all 6 piglets (3 barrows and 3 gilts) in each pen, since one treatment contained 8 pens, n = 8 indicating the data was analyzed in 8 pens per treatment. Please refer to the Line 125: About 1 kg fresh fecal samples from all piglets in each pen (n = 8) were collected.
Comment: About text from L123-132 and variables in Table 3: Authors should think about the most logical order in which to arrange the variables, and then describe the analyses in the same order. For example, organic matter should be mentioned before crude protein and ether extract, as these fractions are included in the former. Dry matter is described earlier in the text, but appears later in the table… Which is Gr (L126)? Chromium (Cr)? This one would be analyzed to determine the digestibility coefficients (L132-134), however other analyses, such as fiber and energy, were described previously. Please, check it.
Response: Thanks for the suggestion. We have revised according to the comment, please refer to Line 127-136: The dried fecal samples and feed were ground and passed through a 1-mm sieve to measure gross energy (GE), dry matter (DM), ash, organic matter (DM-ash, OM), crude protein (CP), ether extract (EE) and chromium (Cr) following the methods of AOAC [22]. The GE content in the dried fecal samples and feed was measure by an automatic isoperibolic oxygen bomb calorimeter (Parr 1281 Automatic Energy Analyzer, Moline, IL, USA). According to Van Soest et al. [23], the fiber analyzer (Ankom Technology, Macedon, NY, USA) was used to measure the level of acid detergent fiber (ADF) and neutral detergent fiber (NDF). The Gr content of the dried fecal samples and feed was measured by the atomic absorption spectrophotometer (Z-5000; Hitachi, Tokyo, Japan).
Comment: L132: Please, replace (Parr 1281, Automatic Energy Analyzer; Moline, IL) by (Parr 1281 Automatic Energy Analyzer, Moline, IL, USA)
Response: Thanks for the suggestion. We have revised according to the comment. Please refer to Line 132.
Comment: L140: Please, replace, Texa, USA by Montgomery, TX, USA
Response: Thanks for the suggestion. We have revised according to the comment. Please refer to Line 144.
Comment: Please, check the order of antioxidant variables in the text is similar and in accordance with the presentation of the data in the table 5.
Response: Thanks for the suggestion. We have revised according to the comment, please refer to Line 145-146.
Comment: Avoid starting sentences with an acronymus or similar (i.e., L148, L157…)
Response: Revised. We have revised according to the comments. Please refer to Line 152: The VFA contents in fecal samples were measured by a Hewlett Packard 5890 gas chromatograph (HP, Pennsylvania) following the procedure mentioned by Long et al. [21]. And also refer to Line 162: Line 157: The 16S rRNA gene of the V3-V4 region was amplified.
Comment: L149: Please, replace Pennsylvania by Avondale, PA, USA
Response: Thanks for the suggestion. We have revised according to the comment, please refer to Line 153.
Comment: L151: Used the gas chromatograph sample bottle to transfer the supernatant. Please, review English editing.
Response: Thanks for the suggestion. We have revised according to the comment, please refer to Line 156-157: The supernatant (1 mL) was transferred into a Gas Chromatograph sample bottle.
Comment: L157: Please, replace nanodrop2000 by NanoDrop 2000 Spectrophotometer, and adding the manufacturer
Response: Revised. Please refer to Line 160-161: The concentration and purity of DNA were detected by NanoDrop 2000 Spectrophotometer (Thermo Scientific, DE, Wilmington, USA).
Comment: L160: Please replace axygen Biosciences, CA, USA by Axygen Biosciences, Union City, CA, USA
Response: Revised. Please refer to Line 165.
Comment: L 161: Please replace Waltham, USA by Waltham, MA, USA
Response: Revised. Please refer to Line 166.
Comment: L163: Please replace San Diego, USA by San Diego, CA, USA
Response: Revised. Please refer to Line 168.
Comment: L165: Trimmatic or Trimmomatic???. Please, check it. Authors described the version used of this software, but no others such as flash software or upars software. Please, be consistent.
Response: Thanks for the useful suggestion. We have added the version of software and revised this sentence into: The original sequencing sequence was quality controlled by Trimmomatic (version 3.29). Flash (version 1.2.7) software was spliced, and UPARSE (version 7.1) software was used to cluster operational taxonomic unit (OTU). Please refer to Line 168-171.
Comment: L155-169: On the other hand, I’m not sure so I’m not an expertise in sequence analyses if some acronymus and/or units should be extended or not in this paragraph. For example, BP (base pair), RDP (Ribosomal Database Project), and QIIME (Quantitative Insights Into Microbial Ecology).
Response: Revised, we have added all the extended names for BP, RDP, QIIME, etc. Please refer to Line 168, 172 and 173.
Comment: Change the references to Table 6 (L237) and Table 7 (L241) by 7 and 8, respectively.
Response: Revised. Please refer to Line 237 and Line 241
Comment: Finally, and taking into account that the references are numbered consecutively, it may not be necessary to differentiate publications of the same author and year with a and b. For example, references number 2 and 3, or 12 and 43. Please, check it according to the journal guidelines.
Response: Revised, we have deleted a and b in the references (Reference 2, 3, 12 and 43). We have checked all the parts according to the journal guidelines.
